# Psychological distress and its associated factors among cancer patients in Nepal: A cross-sectional study

Ankit Acharya[1,2]*, Princy Bhatta[1,2], Dharma Dev Bhatta[3], Murari Man Shrestha[4], Vishnu Prasad Sapkota[5]

1 Department of Public Health, Nobel College, Pokhara University, Kathmandu, Nepal, 2 Research Division, Sustainable Public Health, Bhaktapur, Nepal, 3 Department of Economics, Aishwarya Multiple Campus, Dhangadhi, Nepal, 4 Prevention Oncology Division, Nepal Cancer Hospital and Research Center, Lalitpur, Nepal, 5 Department of Economics, Nepal Commerce Campus, Tribhuvan University, Kathmandu, Nepal

* theankitacharya@gmail.com

## Abstract

Psychological distress, encompassing depression, anxiety, and stress, is common among individuals with cancer but remains inadequately recognized in many low- and middle-income countries, including Nepal. Despite growing emphasis on holistic cancer care, psychological aspects are often overshadowed by clinical and financial concerns. This study aimed to estimate the prevalence of psychological distress among cancer patients in Nepal and to identify its associated sociodemographic, clinical, and economic factors. A cross-sectional study was conducted among cancer patients in two tertiary level referral hospitals in Nepal. The validated Nepali version of the Depression, Anxiety and Stress Scale (DASS-21) was used to assess psychological distress. A total of 262 participants were enrolled. SPSS version 27 was used to perform cleaning, coding and analyses. Associations were examined using binary logistic regression, and adjusted odds ratios were estimated using multivariate logistic regression, with p-values <0.05 considered statistically significant. More than two-thirds of cancer patients had depression (66.8%) and anxiety (68.7%) symptoms, and nearly three-fifths had symptoms of stress (58.0%). Higher odds of psychological distress were observed among participants who were older, unemployed, had fewer years of education, and had advanced-stage cancer. Compared with patients diagnosed at stage I, those diagnosed at II, III, and IV had significantly higher odds of depressive symptoms, with crude odds ratios of 2.75 (95% CI: 1.39-5.42), 2.81 (95% CI: 1.40-5.64), and 4.21 (95% CI: 1.39-12.76), respectively. Although households of cancer patients experienced out of pocket expense, catastrophic health expense and impoverishment, no significant association was found between these indicators and psychological distress. Psychological distress is highly prevalent among Nepali cancer patients and is linked primarily to sociodemographic and clinical factors rather than direct economic burden. Integrating routine mental health screening and basic

**Data availability statement:** All relevant data are made fully available, and included within the manuscript and its Supporting Information files.

**Funding:** The author(s) received no specific funding for this work.

**Competing interests:** The authors have declared that no competing interests exist.

counseling into oncology services is urgently needed to improve patients' overall well-being.

## Introduction

Cancer, one of the deadliest diseases in the history of humankind, was responsible for nearly 20 million new cases and 10 million deaths in 2022 [1,2]. The increasing rates of cancer are evidence that the global disease burden has shifted to non-communicable diseases (NCDs), along with significant inequalities across countries according to the Human Development Index [3]. In 2022, the Global Cancer Observatory reported more than 20,000 new cases of cancer and over 14,000 deaths in Nepal [4]. Increasing cases of cancer in Nepal add challenges to the health system due to the lack of established mechanisms for screening, diagnosis, and treatment for its citizens, and inadequate financing mechanisms to help patients cope with the disease.

The consequences of cancer are not only limited to physical hardships but also financial, psychological, social, and spiritual [5]. Diagnosis of cancer is often associated with fear of the six D's: dependency on family members and medical personnel, disruption of relationships, disfigurement and doubt in self-image, discomfort or pain, disability, and death [6–8]. While moderate fear can motivate adaptive coping and adherence to treatment, excessive fear may overwhelm patients' coping capacity, contributing to maladaptive behaviors and heightened psychological distress, including depression, anxiety, and stress [9]. In addition, cancer diagnosis can negatively affect the sense of coherence (SOC) of patients thus making them less resilient in understanding, dealing, and managing stresses of life [10,11]. Psychological distress is often exaggerated by out-of-pocket expense (OOPE- medical expenses paid directly by patients or their families), catastrophic healthcare expenditure (CATA- occurs when a household's total out-of-pocket health payments equal or exceed 40% of household's capacity to pay), impoverishment (occurs when total household expenditure is smaller than its subsistence spending) due to cancer management, inadequate coping resources, and deleterious coping mechanisms such as delaying or skipping medical care [12]. It is also influenced by side effects of treatment measures such as Radiotherapy [13–15], and Chemotherapy [7,16–18].

Despite substantial global evidence highlighting the unmet psychosocial needs of individuals with cancer [19–21], 40–90% of patients with psychological distress remain unrecognized by clinicians, leading to inadequate care [22–24]. A recent systematic review of cancer populations in Southeast Asia found that the prevalence of anxiety ranged from 7 to 88 percent and depression from 3 to 65 percent, with most studies documenting moderate to severe distress but limited screening and counselling integration in clinical practice [7]. Moreover, while global research increasingly conceptualizes "financial toxicity" as a determinant of psychological well-being [12,25], no study in Nepal has systematically analyzed how financial protection indicators such as OOPE, CATA, and impoverishment relate to psychological distress among cancer patients.

The literature offers several theoretical and conceptual frameworks that explain how socio-demographic, clinical and economic factors are associated with psychological distress. First, the Engel's biopsychosocial model [26] and the WHO's conceptual framework for action on the social determinants of health [27] jointly recognize that health and illness arise from the interplay of biological, psychological, and social factors shaped by wider social and economic conditions. Second, Zafar and Abernethy's financial-toxicity framework [28] conceptualizes the economic burden of cancer as a multidimensional phenomenon encompassing objective financial hardship (e.g., out-of-pocket expenditures, catastrophic spending) and subjective financial distress (e.g., worry, fear of treatment costs, perceived loss of control). Financial hardship operates as both a chronic economic stressor and a psychological stress amplifier, interacting with social and cultural factors to shape patient well-being. Third, the Lazarus and Folkman's psychological stress and coping theory [29] suggests that, psychological distress arises from a dynamic appraisal process: patients evaluate the meaning of a health threat (primary appraisal) and their resources to manage it (secondary appraisal). These appraisals determine emotional and behavioral responses, which may facilitate or hinder adaptation. This theory highlights that, coping strategies (prob-lem-focused, emotion-focused, and meaning-focused) mediate the relationship between perceived stress and psychoso-cial outcomes such as depression, anxiety, and stress. Fourth, evidence from the family-stress and financial-counselling literature suggests the intervening role of social support as a buffering mechanism that mitigates the effects of financial hardship on perceived distress [30]. In low- and middle-income settings such as Nepal, collective family decision-making, shared financing, and cultural norms of resilience may further moderate this relationship, consistent with Wheaton's inter-vening social-support model [31]. Finally, Jones et al.'s recently refined conceptual model of financial hardship after cancer captures four intertwined dimensions- financial coping behaviors, financial consequences, financial worry, and financial depression- that shape patients' experience, and highlights that financial hardship extends beyond medical costs to encompass the psychological and behavioral burdens of managing cancer-related expenses  [32]. Collectively, these perspectives inform a coherent basis for understanding the psychosocial pathways through which these factors influence psychological distress among cancer patients.

Existing literature in Nepal has either focused primarily on estimating financial burden across NCDs, including cancer [33–35], or has explored psychological distress without integrating indicators of financial protection [36–39]. This study therefore fills an important gap by providing the first evidence linking financial hardship and psychological well-being in the context of cancer care in Nepal. By integrating psycho-social and economic perspectives within a low-resource health system context, this study contributes new evidence to the growing international effort to embed mental-health screening and financial protection within comprehensive, person-centered cancer care frameworks.

## Materials and methods

### Ethics statement

Ethical clearance was obtained from the Institutional Review Committee at Nobel College, Pokhara University, Nepal (Ref. No. 079/080/486). Before data collection, a letter from the Department of Public Health, Nobel College was provided to the study sites. After the completion of data collection, a letter of data collection completion was obtained from each study sites.

This study was conducted in accordance with the principles of the Declaration of Helsinki. Written informed consent was obtained from all participants following explanation of the information sheet which included the study's objectives, potential harm and benefits of involvement, approximate time required for the interview, confidentiality, anonymity of personal data obtained, and their right to withdraw their participation from the interview in case they feel uncomfortable. Interviews were conducted either in hospital rooms or in designated spaces specifically allocated for data collection by the nursing in-charge. Patients with critical health conditions were interviewed at the bedside, while those able to participate in exit interviews were interviewed in private spaces near hospital ward, day care units or outpatient areas to ensure confi-dentiality and privacy.

## Study design, respondents, and setting

An institution-based cross-sectional study was conducted from 28 March 2023 to 9 May 2023 in Bhaktapur Cancer Hospital (BCH), a public tertiary-level hospital in Bhaktapur district and from 18th April 2023–13th May 2023 in Nepal Cancer Hospital and Research Center (NCHRC), a private tertiary-level hospital in Lalitpur district in Nepal. These referral-level hospitals serve as central hubs for cancer diagnosis, treatment, and follow-up, attracting patients across all cancer stages and with varying level of financial burden, including patients referred from rural and non-tertiary settings and diverse socio-economic backgrounds. Participants were eligible for inclusion if they had been diagnosed with cancer, were receiving treatment, and had undergone investigatory and therapeutic procedures but had not yet been declared cancer-free at the time of data collection. Exclusion criteria included being younger than 18 years, being critically ill, or being unable to provide the consent or the information required for the study.

## Sampling frame, sample size, and sampling procedure

A total of 400 cancer patients were recruited from selected tertiary-level hospitals in Nepal as part of a larger study investigating the economic burden and psychological impacts of cancer. This sample size was calculated to detect small differences between subgroups (public and private hospitals) (effect size = 0.2) with 80% statistical power and a 5% level of significance [40]. The details of the sample size calculation are provided in the supplementary material (S2 Text).

At each site, participants were selected using a two-stage random sampling approach. First, a sampling frame of all operational dates within the study period was created: 51 consecutive dates from 28 March to 17 May 2023 for BCH, and 28 dates from 18 April to 15 May 2023 for NCHRC. From these frames, 17 and 13 dates, respectively, were randomly selected without replacement using the sample() function in R (seed = 1 for BCH, seed = 2 for NCHRC) (S1 and S2 Tables). On the selected dates, all eligible cancer patients attending outpatient, inpatient, or daycare services between 08:00 and 18:00 were invited to participate, in collaboration with the nurses in charge, ensuring that patient inclusion did not depend on staff discretion. Weekends and public holidays were included to capture the full spectrum of patient visits.

For this specific analysis focused on psychological distress, no stratification by facility type was performed. Of the 400 participants enrolled, 262 consented to and completed the Depression, Anxiety and Stress Scale - 21 Items (DASS-21) assessment tool for measuring psychological distress. The exclusion of non-respondents may have introduced non-response bias, which is acknowledged as a limitation. Analyses of economic outcomes from the broader study, using the same sample, will be reported separately [41].

## Data collection technique and tools

A validated Nepali version of the DASS-21 tool was used to assess psychological distress along three psychological dimensions: depression, anxiety, and stress, which reported good internal consistency, with Cronbach's alpha values of 0.77 for depression (DASS-D), 0.80 for anxiety (DASS-A), and 0.82 for stress (DASS-S) [42]. Economic variables like OOPE, CATA, and impoverishment were calculated based on WHO's distribution of health payments and catastrophic expenditures methodology [43]. Data were collected through face-to-face interviews conducted by the principal investigator (PI) and a team of enumerators. Enumerators were undergraduate public-health students who received a four-hour virtual orientation from the PI on research ethics and the standardized administration of the DASS-21 instrument. Although the training period was brief, the PI's active involvement in the field, continuous supervision, and regular data audits likely mitigated any potential effects on data consistency and ensured the completeness and accuracy of responses.

## Data management and analysis

Raw data was entered in SPSS version 27 for cleaning, coding, and analyses. Descriptive statistics like frequency, proportion, and mean were computed as necessary and presented using tables, figures, and texts. Given the cross-sectional

design of this study, analyses were conducted to identify statistical associations rather than infer causal relationships between variables. In model I, a binary logistic regression was used to calculate the unadjusted odds ratio to examine the association between the dependent and independent variables. In Model II, variables that demonstrated associations at $p < 0.05$ in the bivariate analysis were entered into the multivariate logistic regression model. Statistical significance was defined as a two-sided $p < 0.05$.

### Selection of study variables

The selection of sociodemographic, clinical, and economic variables was guided by established theoretical frameworks and prior empirical research in psycho-oncology and financial-toxicity studies. Sociodemographic variables included age, sex, caste or ethnicity, marital status, education, occupation before and after diagnosis, household size, number of economically active household members, and monthly household income. These variables were included because demographic disadvantage, social position, and limited household resources are consistently linked to greater vulnerability to psychological distress among individuals with cancer [44–50]. Clinical variables comprised cancer type (breast, cervical, lung, prostate, and others), duration since diagnosis, stage of cancer at diagnosis, treatment modalities received (chemotherapy, radiotherapy, surgery, hormonal, supportive or palliative, and others), number of chemotherapy and radiotherapy cycles completed, presence of comorbid chronic disease, type of health facility utilized (public or private), and number of facilities visited for cancer management. These variables were included because disease site, illness duration, stage, and treatment burden are recognized predictors of depression, anxiety, and stress [51–54]. Economic variables encompassed participation in the National Health Insurance Program, receipt of government or provincial subsidies, OOPE, CATA, and household impoverishment. These indicators align with WHO and World Bank endorsed measures of financial protection and capture domains of financial hardship associated with psychological distress [55–58].

### Operational definitions of variables

**DASS-21:** Refers to depression, anxiety, and stress scale, which consists of 21 items. Detailed scoring procedures and cut-off values for each subscale are provided as supplementary files (S3 Text).
**OOPE:** Expense incurred by cancer patients or their family members for medical care not covered by insurance or any form of financial subsidies.
**CATA:** Catastrophic heath expenditure occurred when a household's total out-of-pocket health payments equal or exceed 40% of household's capacity to pay or non-subsistence spending [59].
**Impoverishment:** Occurred when total household expenditure was smaller than its subsistence spending. The subsistence spending per capita, which is also called the poverty line, was calculated to be NRs. 15718.60 in this study.

## Results

In this study, a total of 262 cancer patients gave consent and participated in face-to-face interviews, of whom 143 were recruited from BCH and 119 from NHCRC. The mean age of cancer patients was 51.59 years with a standard deviation of 14.41 years. Nearly one-half of the cancer patients (48.1%) were between the ages of 40–59 years, nearly one-third were 60 years or above, and the remaining were below 40. Nearly two-thirds of the cancer patients were female (63.0%). Cancer patients from Janajati ethnic composition comprised the largest proportion (43.1%), followed by Brahman (30.9%), Chhetri (18.3%), and others (7.7%) respectively. The majority of the cancer patients were married (83.2%). Nearly three-fifths of the cancer patients (58.4%) were employed before cancer diagnosis. The majority of the cancer patients reported being unemployed (76.7%) after the diagnosis of cancer. One-half of the cancer patients had only up to 5 years of education (50%), followed by more than one-third with 6–12 years of education (34.7%). More than one-half (51.5%) had medium family size with 5–8 members, followed by small family size with 4 or less members (40.5%). More than one-third

of cancer patients' households had one (36.6%) and two (36.6%) economically active members. The mean household income of the cancer patients was NRs. 61,469.47, with a standard deviation of NRs. 51,954.42. More than two-fifths of the households had monthly income up to NRS. 50,000. (Table 1)

**Table 1. Socio-demographic characteristics of cancer patients. n = 262.**

| Variables | Frequency |
|---|---|
| | n (%) |
| **Age (in years)** | |
| Mean (SD) | 51.59 (14.41) |
| Less than 40 | 50 (19.1) |
| 40-59 | 126 (48.1) |
| 60 and above | 86 (32.8) |
| **Sex** | |
| Male | 97 (37.0) |
| Female | 165 (63.0) |
| **Caste or ethnic composition** | |
| Brahman | 81 (30.9) |
| Chhetri | 48 (18.3) |
| Janajati | 113 (43.1) |
| Others | 20 (7.6) |
| **Marital status** | |
| Single | 25 (9.5) |
| Married | 218 (83.2) |
| Widowed | 19 (7.3) |
| **Occupation before cancer diagnosis** | |
| Unemployed | 109 (41.6) |
| Employed | 153 (58.4) |
| **Current occupation** | |
| Unemployed | 201 (76.7) |
| Employed | 61 (23.3) |
| **Years of education:** | |
| Up to 5 years | 131 (50.0) |
| 6-12 years | 91 (34.7) |
| More than 12 years | 40 (15.3) |
| **Size of the household/ Family size** | |
| 1 to 4 members (Small) | 106 (40.5) |
| 5 to 8 members (Medium) | 135 (51.5) |
| 9 and more members (Large) | 21 (8.0) |
| **Economically active members in the household (members with income source)** | |
| None | 8 (3.1) |
| One | 96 (36.6) |
| Two | 96 (36.6) |
| Three or more | 62 (23.7) |
| **Household monthly income (NRs.)** | |
| Mean (SD) | |
| Up to 50000 | 159 (60.7) |
| Above 50000 | 103 (39.3) |

The most common cancers identified were breast cancer (22.5%), followed by lung cancer (16.0%), and cervical cancer (14.5%). The majority of the cancer patients (70.6%) were diagnosed with cancer within the last 12 months. At the time of diagnosis, 22.5% of patients were diagnosed at stage I (n = 59), 35.9% at stage II (n = 94), 32.4% at stage III (n = 85), and 9.2% at stage IV (n = 24). In terms of treatment, the majority of patients received chemotherapy (84.7%, n = 222), followed by surgery (19.8%, n = 52) and radiotherapy (19.5%, n = 51). Smaller proportions received hormonal therapy (1.1%, n = 3), supportive or palliative treatment (7.3%, n = 19), or other forms of treatment (4.6%, n = 12), while 6.1% (n = 16) did not receive any treatment. The mean number of chemotherapy cycles completed was 6.63 (SD = 5.01), while the mean number of radiotherapy fractions was 15.71 (SD = 11.41). The majority of the cancer patients had no other chronic disease (82.4%). Nearly two out of five cancer patients were insured under the National Health Insurance Program of Nepal. The majority of cancer patients (92.0%) received subsidies (subsidies for medical treatment of deprived citizens or provincial subsidies) for cancer management. Among all, nearly three-fifths of the cancer patients (58.0%) reported OOPE for cancer management. Among 242 respondents, who provided data for CATA, nearly half (47.4%) incurred catastrophic healthcare expenditure due to cancer management. Unlike the analysis of CATA, which required data on total household consumption, food expenditure (to estimate subsistence spending), and OOPE, the assessment of impoverishment additionally required comprehensive pre- and post-payment household expenditure information across all major categories defined by the Classification of Individual Consumption According to Purpose including food and non-food expenses, to identify movements across the poverty threshold. As this analysis necessitated complete and itemized consumption records, only 177 respondents provided sufficiently detailed data, resulting in a smaller analytic sample. Among them, more than two-fifths (40.1%) of the cancer patients were dragged below the poverty line due to cancer management. Nearly one-half of the cancer patients (45.4%) received cancer management services from private hospitals. With regard to health facility use, 26.7% of patients (n = 70) sought care from a single facility, 32.4% (n = 85) from two facilities, 22.9% (n = 60) from three facilities, and 17.9% (n = 47) from four or more facilities. (Table 2)

More than two-thirds of cancer patients who reported to the DASS-21 tool reported symptoms of depression (66.8%) and anxiety (68.7%), and nearly three-fifths reported symptoms of stress (58.0%). Higher levels of depression, anxiety, and stress were associated with more advanced stages of cancer. (Fig 1)

Table 3 presents unadjusted (Model I) and adjusted (Model II) odds ratios for socio-demographic and clinical factors associated with depressive symptoms among cancer patients. In Model I, significant predictors of depressive symptoms included: age, caste or ethnic composition, occupation, years of education, number of economically active members in the family, household monthly income, and stage of cancer at the time of diagnosis. Patients aged 60 years or above had higher odds of having depressive symptoms compared with those under 40 years (cOR 2.33; 95% CI 1.12, 4.86; p-value = 0.023). Compared with patients from a Brahman background, those from Chhetri (cOR 2.97; 95% CI 1.33–6.63; p-value = 0.008), Janajati (cOR 2.05; 95% CI 1.13–3.72; p-value = 0.017), and other caste or ethnic groups (cOR 3.54; 95% CI 1.09–11.50; p-value = 0.036) had significantly higher odds of reporting depressive symptoms. Patients who were unemployed following their cancer diagnosis had significantly higher odds of reporting depressive symptoms compared with those who remained employed (cOR 1.87; 95% CI 1.04, 3.36; p-value = 0.038). Lower education showed a strong association, with those having ≤5 years of schooling nearly four times more likely to report depressive symptoms than those with >12 years (cOR 3.94; 95% CI 1.88, 8.28; p-value < 0.001). In reference to those who had 3 or more economically active members in the family, those with only one had 2.03 higher odds of having depressive symptoms. Similarly, households with income ≤NRs. 50,000 had twice the odds of depressive symptoms compared with higher-income households (cOR 2.01; 95% CI 1.19, 3.40; p-value = 0.009). Advanced cancer stages also showed progressively higher odds, with fourth-stage patients having over four times the odds compared with those at stage one (cOR 4.21; 95% CI 1.39, 12.76; p-value = 0.011). In Model II, variables that were significant in Model I were simultaneously included to account for potential confounding and identify independent predictors of depression. In Model II, the associations for caste/ethnicity, education, income, and stage of cancer remained significant, while the effects of age, occupation, and number

**Table 2. Clinical characteristics of the patients, health services used, payment mechanism, experience of financial hardship, and coping strategies adopted by the households. n = 262.**

| | Frequency n (%) |
|---|---|
| **Cancer diagnosed** | |
| Breast Cancer | 59 (22.5) |
| Cervical Cancer | 38 (14.5) |
| Lung Cancer | 42 (16.0) |
| Prostate Cancer | 14 (5.3) |
| Others* | 109 (41.6) |
| **Duration since cancer diagnosis** | |
| More than 2 years | 25 (9.5) |
| 1-2 years | 52 (19.8) |
| 6-12 months | 77 (29.4) |
| Less than 6 months | 108 (41.2) |
| **Stage of cancer at the time of diagnosis** | |
| First | 59 (22.5) |
| Second | 94 (35.9) |
| Third | 85 (32.4) |
| Fourth | 24 (9.2) |
| **Treatment received (Multiple response)** | |
| Chemotherapy (222) | 222 (84.7) |
| Radiotherapy (211) | 51 (19.5) |
| Surgery (262) | 52 (19.8) |
| Hormonal therapy (262) | 3 (1.1) |
| Supportive/ palliative treatment (262) | 19 (7.3) |
| Others (262) | 12 (4.6) |
| None (262) | 16 (6.1) |
| **Number of treatment cycles completed for chemotherapy (n = 222)** | |
| Mean (SD) | 6.63 (5.01) |
| 1-3 cycles | 73 (32.9) |
| 4-6 cycles | 57 (25.7) |
| 7-9 cycles | 58 (26.1) |
| 10 or more cycles | 34 (15.3) |
| **Number of treatment cycles completed for radiotherapy (n = 51)** | |
| Mean (SD) | 15.71 (11.41) |
| 1-10 fractions | 25 (49.0) |
| 11-20 fractions | 7 (13.7) |
| More than 20 fractions | 19 (37.3) |
| **Presence of any other chronic disease** | |
| No | 216 (82.4) |
| Yes | 46 (17.6) |
| **Insurance under NHIP** | |
| Insured | 101 (38.5) |
| Not insured | 161 (61.5) |
| **Received subsidies (Bipanna Nagarik Kosh and/or Provincial subsidies)** | |
| No | 21 (8.0) |
| Yes | 241 (92.0) |

*(Continued)*

**Table 2.** (Continued)

| | Frequency |
| --- | --- |
| | n (%) |
| **OOPE** | |
| No | 110 (42.0) |
| Yes | 152 (58.0) |
| **CATA (242)** | |
| No | 122 (52.6) |
| Yes | 110 (47.4) |
| **Impoverishment (177)** | |
| No | 106 (59.9) |
| Yes | 71 (40.1) |
| **Type of health facility visited** | |
| Public | 143 (54.6) |
| Private | 119 (45.4) |
| **Number of health facilities visited for cancer management** | |
| One | 70 (26.7) |
| Two | 85 (32.4) |
| Three | 60 (22.9) |
| Four or more | 47 (17.9) |

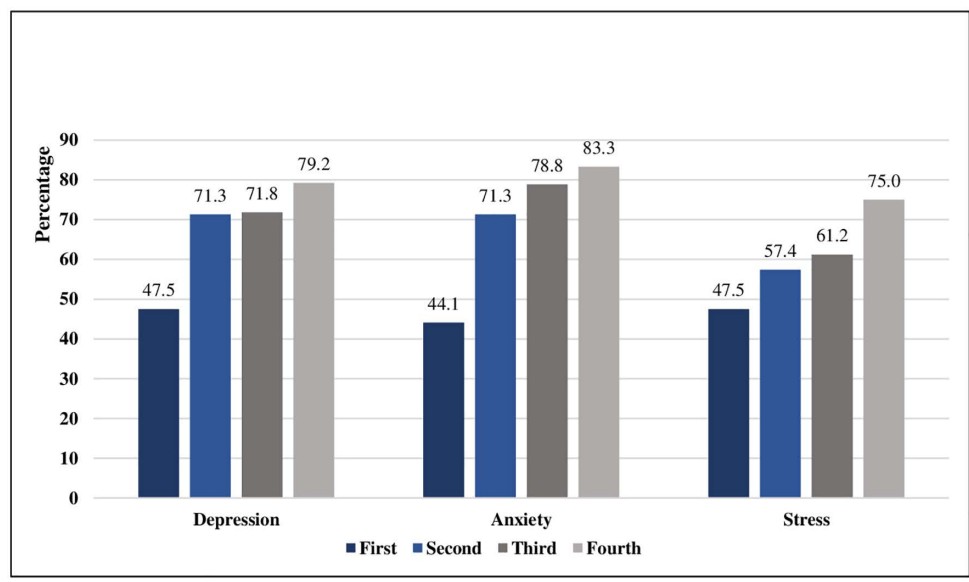

**Fig 1. Comparison of depression, anxiety and stress across stages of cancer diagnosis.**

of economically active members were attenuated. For example, ≤ 5 years of education (aOR 4.06; 95% CI 1.61, 10.20; p-value = 0.003) and income ≤NRs. 50,000 (aOR 2.12; 95% CI 1.09, 4.14 p-value = 0.028) continued to demonstrate strong associations. The association between advanced cancer stages and depression also persisted, with stage IV patients having the highest adjusted odds (aOR 4.76; 95% CI 1.33, 17.03; p-value = 0.017). (Table 3)

**Table 3. Bivariate and multivariate analysis for socio-demographic and clinical characteristic variables with depressive symptoms.**

| Patients' characteristics | Depressive symptoms | | Model I, | p-value | Model II, | p-value |
|---|---|---|---|---|---|---|
| | Yes | No | cOR (95%CI) | | aOR (95%CI) | |
| | n (%) | n (%) | | | | |
| **Age (in years) Mean (SD): 53.61 (14.41)** | | | | | | |
| Less than 40 (50) | 27 (54.0) | 23 (46.0) | Ref | Ref | Ref | Ref |
| 40-59 (126) | 85 (67.5) | 41 (32.5) | 1.77 (0.90, 3.45) | 0.096 | 1.14 (0.48, 2.68) | 0.766 |
| 60 and above (86) | 63 (73.3) | 23 (26.7) | **2.33 (1.12, 4.86)** | **0.023** | 1.40 (0.55, 3.54) | 0.483 |
| **Sex** | | | | | | |
| Male (97) | 59 (69.8) | 38 (39.2) | Ref | Ref | | |
| Female (165) | 116 (70.3) | 49 (29.7) | 1.53 (0.90, 2.58) | 0.117 | | |
| **Marital status** | | | | | | |
| Single (25) | 13 (52.0) | 12 (48.0) | Ref | Ref | | |
| Married (218) | 147 (67.4) | 71 (32.6) | 1.91 (0.83, 4.40) | 0.289 | | |
| Widowed (19) | 15 (78.9) | 4 (21.1) | 3.46 (0.89, 13.40) | 0.552 | | |
| **Occupation status after cancer diagnosis** | | | | | | |
| Employed (61) | 34 (55.7) | 27 (44.3) | Ref | Ref | | |
| Unemployed (201) | 141 (70.1) | 60 (29.9) | **1.87 (1.04, 3.36)** | **0.038** | | |
| **Level of education** | | | | | | |
| Up to 5 years (131) | 100 (76.3) | 31 (23.7) | **3.94 (1.88, 8.28)** | **< 0.001** | **4.06 (1.61, 10.20)** | **0.003** |
| 6-12 years (91) | 57 (62.6) | 34 (37.4) | 2.05 (0.96, 4.36) | 0.062 | 2.35 (0.97, 5.73) | 0.060 |
| More than 12 years (40) | 18 (45.0) | 22 (55.0) | Ref | Ref | Ref | Ref |
| **Size of the household/ Family size** | | | | | | |
| 1 to 4 members (Small) (106) | 65 (61.3) | 41 (38.7) | Ref | Ref | | |
| 5 to 8 members (Medium) (135) | 95 (70.4) | 40 (29.6) | 1.50 (0.88, 2.57) | 0.383 | | |
| 9 and more members (Large) (21) | 15 (71.4) | 6 (28.6) | 1.58 (0.57, 4.39) | 0.921 | | |
| **Economically active members in the household (members with income source)** | | | | | | |
| None (8) | 3 (37.5) | 5 (62.5) | 0.41 (0.09, 1.85) | 0.244 | 0.19 (0.03, 1.07) | 0.060 |
| One (96) | 72 (75.0) | 24 (25.0) | **2.03 (1.02, 4.03)** | **0.044** | 1.47 (0.63, 3.43) | 0.369 |
| Two (96) | 63 (65.6) | 33 (34.4) | 1.29 (0.67, 2.49) | 0.449 | 1.24 (0.59, 2.62) | 0.570 |
| Three or more (62) | 37 (59.7) | 25 (40.3) | Ref | Ref | Ref | Ref |
| **Household monthly income (NRs.) Mean (SD)** | | | | | | |
| Up to 50000 (159) | 116 (73.0) | 43 (27.0) | **2.01 (1.19, 3.40)** | **0.009** | **2.12 (1.09, 4.14)** | **0.028** |
| Above 50000 (103) | 59 (57.3) | 44 (42.7) | Ref | Ref | Ref | Ref |
| **Cancer diagnosed** | | | | | | |
| Breast Cancer (59) | 41 (69.5) | 18 (30.5) | 1.54 (0.79, 3.02) | 0.208 | | |
| Cervical Cancer (38) | 28 (73.7) | 10 (26.3) | 1.90 (0.84, 4.29) | 0.125 | | |
| Lung Cancer (42) | 30 (71.4) | 12 (28.6) | 1.69 (0.78, 3.66) | 0.181 | | |
| Prostate Cancer (14) | 11 (78.6) | 3 (21.4) | 2.48 (0.66, 9.41) | 0.181 | | |
| Others* (109) | 65 (59.6) | 44 (40.4) | Ref | Ref | | |
| **Duration since cancer diagnosis** | | | | | | |
| More than 2 years (25) | 17 (68.0) | 8 (32.0) | 1.20 (0.48, 3.04) | 0.699 | | |
| 1-2 years (52) | 38 (73.1) | 14 (26.9) | 1.53 (0.74, 3.18) | 0.249 | | |
| 6-12 months (77) | 51 (66.2) | 26 (33.8) | 1.11 (0.60, 2.05) | 0.742 | | |
| Less than 6 months (108) | 69 (63.9) | 39 (36.1) | Ref | Ref | | |
| **Stage of cancer at the time of diagnosis** | | | | | | |
| First (59) | 28 (47.5) | 31 (52.5) | Ref | Ref | Ref | Ref |
| Second (94) | 67 (71.3) | 27 (28.7) | **2.75 (1.39, 5.42)** | **0.004** | **3.23 (1.52, 6.87)** | **0.002** |

*(Continued)*

**Table 3.** (Continued)

| Patients' characteristics | Depressive symptoms | | Model I, | p-value | Model II, | p-value |
|---|---|---|---|---|---|---|
| | Yes | No | cOR (95%CI) | | aOR (95%CI) | |
| | n (%) | n (%) | | | | |
| Third (85) | 61 (71.8) | 24 (28.2) | **2.81 (1.40, 5.64)** | **0.004** | **2.88 (1.30, 6.41)** | **0.009** |
| Fourth (24) | 19 (79.2) | 5 (20.8) | **4.21 (1.39, 12.76)** | **0.011** | **4.76 (1.33, 17.03)** | **0.017** |
| **Presence of any other chronic disease** | | | | | | |
| No (216) | 146 (67.6) | 70 (32.4) | Ref | Ref | | |
| Yes (46) | 29 (63.0) | 17 (37.0) | 0.82 (0.42, 1.59) | 0.552 | | |
| **Insurance under NHIP** | | | | | | |
| Insured (101) | 69 (68.3) | 32 (31.7) | Ref | Ref | | |
| Not insured (161) | 106 (65.8) | 55 (34.2) | 0.89 (0.53, 1.52) | 0.679 | | |
| **Received subsidies (Bipanna Nagarik Kosh and/or Provincial subsidies)** | | | | | | |
| No (21) | 18 (85.7) | 3 (14.3) | 3.21 (0.92, 11.21) | 0.068 | | |
| Yes (241) | 157 (65.1) | 84 (34.9) | Ref | Ref | | |
| **Type of health facility visited** | | | | | | |
| Public (143) | 98 (68.5) | 45 (31.5) | 0.84 (0.50, 1.41) | 0.513 | | |
| Private (119) | 77 (64.7) | 42 (35.3) | Ref | Ref | | |
| **Number of health facilities visited for cancer management** | | | | | | |
| One (70) | 47 (67.1) | 23 (32.9) | Ref | Ref | | |
| Two (85) | 50 (58.8) | 35 (41.2) | 0.70 (0.36, 1.35) | 0.288 | | |
| Three (60) | 48 (80.0) | 12 (20.0) | 1.96 (0.88, 4.38) | 0.102 | | |
| Four or more (47) | 30 (63.8) | 17 (36.2) | 0.86 (0.40, 1.88) | 0.711 | | |
| **OOPE** | | | | | | |
| No (110) | 79 (71.8) | 31 (28.2) | Ref | Ref | | |
| Yes (152) | 56 (36.8) | 96 (63.2) | 0.67 (0.40, 1.14) | 0.143 | | |
| **CATA (n = 232)** | | | | | | |
| No (122) | 85 (69.7) | 37 (30.3) | Ref | Ref | | |
| Yes (110) | 70 (63.6) | 40 (36.4) | 0.76 (0.44, 1.32) | 0.330 | | |
| **Impoverishment (n = 177)** | | | | | | |
| No (106) | 75 (70.8) | 31 (29.2) | Ref | Ref | | |
| Yes (71) | 44 (62.0) | 27 (38.0) | 0.67 (0.38, 1.27) | 0.223 | | |

S3 Table presents unadjusted (Model I) and adjusted (Model II) odds ratios for socio-demographic and clinical factors associated with anxiety symptoms among cancer patients. There was a significant association of the following variables with anxiety symptoms among cancer patients: age, caste or ethnic composition, occupation, years of education, family size, household monthly income, and stage of cancer at the time of diagnosis. In reference to those who were diagnosed with first stage of cancer, those who were diagnosed with second, third and fourth stage of cancer had 3.15 higher odds (cOR 3.15; 95% CI 1.59, 6.22; p-value < 0.001), 4.72 higher odds (cOR 4.72; 95% CI 2.27, 9.82; p-value < 0.001) and 6.35 higher odds (cOR 6.35; 95% CI 1.93, 20.87; p-value = 0.002) of having anxiety symptoms respectively. In Model II, education up to primary level (aOR 4.42; 95% CI 1.71, 11.47; p-value = 0.002), medium household size (aOR 2.07; 95% CI 1.02, 4.21; p-value = 0.045), low income (aOR 2.29; 95% CI 1.22, 4.31; p-value = 0.010), and advanced cancer stage (second stage: aOR 4.38; 95% CI 1.98, 9.69; p-value < 0.001; third stage: aOR 4.45; 95% CI 1.98, 10.01; p-value < 0.001; fourth stage: aOR 7.53; 95% CI 1.84, 30.87; p-value = 0.005) remained statistically significant. Other variables, including age, sex, occupation, and economically active household members, were not significant in Model II, indicating that their associations were explained by confounding factors. (S3 Table)

S4 Table presents unadjusted (Model I) and adjusted (Model II) odds ratios for socio-demographic and clinical factors associated with stress symptoms among cancer patients. In Model I, older age (≥60 years), marital status (married and widowed), unemployment, lower education (≤5 years), medium family size (5–8 members), and advanced cancer stage (fourth stage) were significantly associated with stress. In Model II, after adjusting, only advanced cancer stage (fourth stage) remained independently associated with stress symptoms, with the strength of association slightly increasing (from cOR 3.32; 95% CI 1.16, 9.55; p-value = 0.026 to aOR 3.47; 95% CI 1.03, 11.66; p-value = 0.044). (S4 Table)

## Discussion

This study found a high prevalence of depression, anxiety, and stress among cancer patients, with more than two-thirds of cancer patients reporting depression and anxiety symptoms. Consistent with our findings, previous studies have reported that almost two-thirds of cancer patients had depression [60,61]. Comparable findings have been reported in other studies, where between one-third and two-thirds of cancer patients experienced depression and anxiety, though some studies noted prevalence rates as high as 90% [36–38,44,50,60–65]. An umbrella review of systematic reviews and meta-analyses reported a summary prevalence of depression among cancer survivors of 33.2% (95% CI: 27.6, 38.7) and anxiety of 30.6% (95% CI: 24.0, 37.1) [66]. This indicates a comparatively greater psychological burden among Nepali cancer patients. With regard to stress, an integrated review reported that approximately 60–70% of cancer patients experience stress, which is broadly consistent with the present study, in which nearly three-fifths of patients reported stress [67]. However, previous studies have shown wide variation, ranging from about one-fifth to nearly four-fifths [36,38,50,64]. Although the Nepali version of the DASS-21 has demonstrated sound psychometric properties (strong reliability and construct validity) among Nepali adults [42], the instrument measures symptom severity rather than clinical diagnosis. Several items such as those reflecting restlessness, breathing difficulty, or loss of initiative can mirror cancer-related physical symptoms, potentially inflating distress scores. In South Asian contexts such as Nepal, limited psychosocial support and persistent uncertainty surrounding disease prognosis, treatment outcomes, and financial stability often leads patients to express emotional distress through physical symptoms such as pain, fatigue, and insomnia, leading to heightened attention to bodily sensations and amplifying overall psychological burden [68]. Consistent with emerging cognitive-affective models, such as the Cancer Threat Interpretation Model, pain and somatic signals may be cognitively appraised as potential signs of recurrence, reinforcing fear, worry, and maladaptive coping [69]. Recent evidence also suggests that somatosensory amplification and fear of cancer recurrence mediate the relationship between bodily awareness and psychological distress [70]. The high prevalence reported in this study may reflect a genuine psychosocial burden shaped by the interplay of cultural expression, stigma surrounding cancer and mental health, pain-related threat appraisal, and scarcity of mental health integration within oncology services.

In this study, participants aged 60 years and older had significantly higher odds of reporting depression, anxiety, and stress than younger individuals, demonstrating a strong association between older age and psychological distress. This finding is similar to findings from several other studies [44]. This may be due to cumulative health burdens, high probability of cancer metastasis, increased financial strain, diminished social support, low sense of coherence, fear of disability and death, and increased feelings of vulnerability and dependency.

The study findings also revealed that psychological distress was notably higher among participants who reported having no occupation after their cancer diagnosis. Lack of a stable income may heighten financial insecurity and psychological dependence, which in turn are associated with reduced self-esteem and greater emotional distress, consistent with evidence that persistent financial hardship can diminish self-esteem and contribute to depressive symptoms [45]. In this study, the proportion of participants reporting unemployment increased from nearly two-fifths before diagnosis to almost four-fifths after diagnosis, based on retrospective self-reports. However, as this information was not derived from longitudinal follow-up, the observed change should be interpreted as descriptive rather than causal. Future studies employing prospective designs is warranted to examine the reasons for the employment loss after diagnosis, and how it impacts their

health and well-being. Prior studies have similarly shown that unemployment and reduced work capacity are common among cancer survivors and are closely linked to psychological distress [46–48].

Educational status was also significantly associated with psychological distress, i.e., the lower the years of education higher the risk of psychological distress. In comparison to those who had more than 12 years of education, the odds of depression, anxiety, and stress symptoms were higher among those who had only up to five years of education. This finding suggests that limited educational attainment may restrict health literacy and coping resources, thereby heightening vulnerability to psychological distress.

Family size was not significantly associated with psychological distress in this study. This might be true because irrespective of the family size, every member of the family comes together to take care of the one who is ill. Another plausible explanation is that psychological distress may be more strongly influenced by the availability of economically active members and quality of social support within the household rather than its absolute size. However, several studies report an increased risk of psychological distress with an increase in household size [49,50]. Future studies should therefore investigate how variations in intra-household roles, caregiving arrangements, and economic contributions influence psychological outcomes among cancer patients in Nepali context. The number of economically active members in the household was significantly associated with psychological distress. In comparison to those households with three or more members with active income sources, those households with only one economically active member had higher odds of depression. In reference to those households with income more than NRs. 50,000, the odds of depression and anxiety were higher among those who had less income.

The stage of cancer at the time of diagnosis was significantly associated with psychological distress. This study found that, in reference to those patients diagnosed with the first stage of cancer, those who were diagnosed with the second, third, and fourth stages had higher odds of depressive, anxiety, and stress symptoms. This result is similar to findings from other studies, which report that second to fourth-stage diagnosis results in increased psychological distress than those diagnosed at the initial stage of cancer [51,52]. This is true in the case of Nepal because a higher stage of cancer diagnosis leads to panicking of entire family members which ultimately affects the psychological well-being of the cancer patient. Also, higher stages of cancer represent the severity of the cancer thus requiring costly treatment services, regular follow-up to manage the disease, and family and social support. At some stage of their lives, cancer patients think of themselves as a burden to the family, thus deteriorating their psychological well-being [53,54].

This study suggests that nearly three-quarters of patients sought care from more than one facility. The use of multiple facilities suggests patterns of care-seeking that may reflect referrals between centers, patients' efforts to access specialized services not available in a single site, or fragmented care pathways that could increase both the financial and psychological burden of cancer management. In the Nepali context, this pattern may reflect patients initially seeking care at nearby facilities before travelling to Kathmandu, Bhaktapur or abroad for specialized oncology services.

Much of the global literature has identified financial burden as a strong correlate of psychological distress [55–58]. Contrary to expectation, no significant association was observed between OOPE, CATA or impoverishment and psychological distress in this study. Several methodological and contextual factors may explain this finding. OOPE data are typically highly skewed, and the large standard errors observed for these variables suggest that residual skewness and heteroskedasticity may have reduced statistical power. Transformations such as log(OOPE + 1) or log(OOPE/total consumption) can stabilize variance but may also introduce interpretational bias and loss of comparability when zeros are present or distributions deviate from log-normality. Because the primary objective of this cross-sectional study was to identify associations between financial indicators and psychological distress rather than to model the distributional properties of expenditure data, logistic regression was used as the most appropriate analytic framework. Future research that seeks to model expenditure magnitudes or jointly analyze costs and psychological outcomes should evaluate these transformations or apply generalized linear models with a log link and gamma distribution to assess the robustness of the findings, as recommended in prior methodological literature [71–73]. Conventional indicators of financial burden capture only objective

expenditures and may neglect the subjective experience of financial toxicity, the perceived cost-related stress that can independently impair psychological well-being among cancer survivors and may therefore account for the absence of association observed in this study. Unmeasured variables such as coping capacity, resilience, health literacy, household debt, and pre-existing mental-health conditions represent additional measurement constraints, as their exclusion may have introduced residual confounding by influencing both financial hardship and psychological distress.

Beyond analytic considerations, real-world contextual factors may also shape the association between financial hardship and psychological distress. Following a cancer diagnosis, the immediate psychological impact of treatment-related expenses may be counterbalanced by structural and social support mechanisms such as government subsidies, health insurance, and informal family financing, particularly in developing countries where collective responsibility, extended-family support, and informal financial networks are central to illness management, and where social connectedness has been shown to foster resilience and emotional recovery among cancer survivors [74–77]. The literature indicates that financial strain, emotional distress, and social and spiritual disruption constitute a broader "psychosocial cost burden" experienced by patients and families [5,78]. Jones and colleagues (2024) argue that multiple moderating factors including socioeconomic status, self-efficacy, and social support can either amplify or attenuate distress [32]. Similarly, Park and Look (2018) observed that although higher financial burden showed a significant association with poorer quality of life and greater psychological distress among cancer patients, its association with depressive symptoms was not significant after adjusting for covariates. This finding suggests that depressive symptoms do not always correspond to the magnitude of financial pressure, possibly because of the already high baseline level of psychological distress [79] or the buffering effect of protective sociocultural influences. In Nepal, culturally grounded spiritual and contemplative practices, meaning-making processes surrounding illness and suffering, cultural norms discouraging open discussion of financial strain- together with strong family and community support- may collectively moderate or mask the psychological impact of economic burden, even when treatment costs are high. The absence of a statistical association in this study should therefore not be interpreted as evidence that financial hardship has no effect on patients' psychological well-being, but as a call for longitudinal, mixed-methods research using context-sensitive instruments to examine these relationships rigorously.

This study has several limitations that should be considered when interpreting the findings. First, the relatively small analytic sample may have reduced statistical power to detect associations and limited the generalizability of results beyond the study sites. The cross-sectional design restricts causal inference between psychological outcomes and explanatory variables. The use of face-to-face interviews may have introduced social desirability bias, as some participants might have under-reported or modified their emotional experiences to align with perceived expectations of the interviewer. Non-response and self-selection bias is also possible, as individuals experiencing greater illness severity or psychological distress may have declined participation owing to fatigue, stigma, or discomfort. As tertiary-level referral centers, the selected study hospitals are more likely to treat patients with advanced disease and higher financial burden, introducing selection bias. Consequently, the elevated levels of psychological distress observed may reflect the intensive treatment demands and substantial financial risk of this population, and the reported prevalence may therefore over- or underestimate the true burden in the broader oncology population. Second, psychological distress was measured using a self-reported screening tool rather than clinical evaluation, which may have introduced measurement and cultural response biases. Moreover, the insignificant associations observed between financial hardship indicators (OOPE, CATA, and impoverishment) and psychological distress may partly reflect methodological limitations. Financial hardship is a multidimensional construct that involves material, psychological, and behavioral responses to economic strain, and inconsistencies in its measurement can substantially influence observed relationships [80]. In this study, financial data were collected at a single cross-sectional time point, which may not have coincided with peak treatment phases or periods of acute economic pressure, thereby underestimating the short-term psychological impact of financial stress [28,80].

## Conclusion

This study found a high prevalence of depression, anxiety, and stress among cancer patients, particularly among older adults, those who were unemployed, with lower educational attainment, or diagnosed at advanced-stage. These findings underline the need for practical and sustainable interventions to address psychological distress as an integral component of cancer care.

In light of our findings, routine mental-health screening can be incorporated into institutional cancer-care pathways to facilitate early identification and management of psychological distress. Oncology nurses and allied health personnel can be trained in basic psychosocial counselling and communication skills to provide immediate support and facilitate appropriate referrals for both patients and their caregivers, recognizing their shared psychological burden throughout the course of illness.

These actions align with the Nepal Cancer Control Strategy (2024–2030), which emphasizes enhanced capacity in diagnosis, treatment, and rehabilitation, and the National Health Financing Strategy (2023–2033), which aims to reduce financial and social vulnerability caused by illness [81,82]. Integrating psychosocial screening and counselling within these national frameworks would help mitigate both the psychological and economic burden of cancer and advance equitable, person-centered oncology care in Nepal.

## Supporting information

**S1 Data. Fully anonymized dataset used for analysis in this study.**
(XLSX)

**S1 Text. Study questionnaires.**
(PDF)

**S2 Text. Sample size calculation.**
(DOCX)

**S3 Text. Operational Definitions and Scoring Criteria.**
(DOCX)

**S1 Text. Abstract in Nepali language.**
(PDF)

**S1 Table. Random sampling for public hospital.**
(PDF)

**S2 Table. Random sampling for private hospital.**
(PDF)

**S3 Table. Bivariate and multivariate for socio-demographic and clinical characteristic variables with anxiety symptoms.**
(DOCX)

**S4 Table. Bivariate and multivariate for socio-demographic and clinical characteristic variables with stress symptoms.**
(DOCX)

## Acknowledgments

We would like to thank Ms. Sushma Gauli and Ms. Smriti Sedhain for their support in data collection at the study sites. We are indebted to cancer patients and their caregivers for providing their time and required information for this study.

## Author contributions

**Conceptualization:** Ankit Acharya, Vishnu Prasad Sapkota.

**Data curation:** Ankit Acharya, Vishnu Prasad Sapkota.

**Formal analysis:** Ankit Acharya, Princy Bhatta, Vishnu Prasad Sapkota.

**Funding acquisition:** Ankit Acharya.

**Investigation:** Ankit Acharya, Princy Bhatta, Murari Man Shrestha.

**Methodology:** Ankit Acharya, Princy Bhatta, Dharma Dev Bhatta, Murari Man Shrestha, Vishnu Prasad Sapkota.

**Project administration:** Ankit Acharya, Princy Bhatta, Murari Man Shrestha.

**Resources:** Ankit Acharya, Murari Man Shrestha.

**Software:** Ankit Acharya.

**Supervision:** Ankit Acharya, Murari Man Shrestha, Vishnu Prasad Sapkota.

**Validation:** Ankit Acharya, Dharma Dev Bhatta, Vishnu Prasad Sapkota.

**Visualization:** Ankit Acharya.

**Writing – original draft:** Ankit Acharya, Princy Bhatta.

**Writing – review & editing:** Ankit Acharya, Dharma Dev Bhatta, Murari Man Shrestha, Vishnu Prasad Sapkota.

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
