## [Decision Letter · Decision Letter 0]

1 Sep 2025

PMEN-D-25-00335

Psychological distress and its associated factors among cancer patients in Nepal: a cross-sectional study

PLOS Mental Health

Dear Dr. Acharya,

Thank you for submitting your manuscript to PLOS Mental Health. After careful consideration, we feel that it has merit but does not fully meet PLOS Mental Health’s publication criteria as it currently stands. Therefore, we invite you to submit a revised version of the manuscript that addresses the points raised during the review process.

We look forward to receiving your revised manuscript.

Kind regards,

Laalithya Konduru

Academic Editor

PLOS Mental Health

Journal Requirements:

1. Please ensure that your Ethics Statement is available in its entirety at the beginning of your Methods section, under a subheading 'Ethics Statement'.

2. We have noticed that you have uploaded Supporting Information files, but you have not included a list of legends. Please add a full list of legends for your Supporting Information files after the references list.

3. We note that there is identifying data in the Supporting Information file “Dataset.xlsx”. Due to the inclusion of these potentially identifying data, we have removed this file from your file inventory. Prior to sharing human research participant data, authors should consult with an ethics committee to ensure data are shared in accordance with participant consent and all applicable local laws.

-Location data

Reviewers' comments:

Reviewer's Responses to Questions

**Comments to the Author**

1. Does this manuscript meet PLOS Mental Health’s publication criteria?

Reviewer #1: Yes

Reviewer #2: Partly

2. Has the statistical analysis been performed appropriately and rigorously?

Reviewer #1: Yes

Reviewer #2: No

3. Have the authors made all data underlying the findings in their manuscript fully available (please refer to the Data Availability Statement at the start of the manuscript PDF file)?

Reviewer #1: Yes

Reviewer #2: Yes

4. Is the manuscript presented in an intelligible fashion and written in standard English?

Reviewer #1: Yes

Reviewer #2: Yes

Reviewer #1: 1. The article provides a clear and data-driven rationale for the study, establishing the growing burden of cancer globally and in Nepal. It effectively contextualizes both the clinical and psychosocial dimensions of cancer, with a specific focus on the intersection of financial hardship and psychological distress which is well articulated and highly relevant, especially for low- and middle-income country contexts where cancer care infrastructure is often limited.

2. While the inclusion of global and national statistics is important, the opening paragraphs could benefit from slight condensation to avoid overwhelming the reader with numbers. Consider prioritizing the most critical figures and integrating them more smoothly into the narrative.

3. The relationship between the "six D’s" and psychological distress could be explained more explicitly. Currently, the transition from physical to psychosocial consequences feels abrupt.

4. Abbreviations like OOPE and CATA are introduced without prior definition. These should be spelled out when first mentioned to ensure clarity for readers unfamiliar with the terms.

5. The challenges in the Nepalese context are noted, but the rationale for focusing on tertiary-level referral hospitals could be further strengthened. Briefly discuss their significance in the healthcare system and why they are a suitable setting for this study.

6. Ensure that all claims—especially those linking treatment side effects to psychological distress—are supported by high-quality, recent references. This will bolster the credibility of the argument.

7. The rationale for analyzing 262 participants (out of 400) is well explained, but consider discussing potential bias introduced by excluding non-DASS-21 respondents (e.g., were there notable differences?). Sampling procedure should be described more clearly. It is not clear how patients were selected from the hospital population—was it random, purposive, or convenience sampling?

8. The sample size formula is included but incomplete. Either present the full derivation with parameter values or simplify and state that it was calculated based on expected effect size and power. P-value should be corrected: “P-values less than 0.05 was considered as significant” should read “were considered significant.”

9. The DASS-21 tool is well-described, but more information on its validation in the Nepali context would enhance credibility (e.g., reliability coefficients, cultural appropriateness). Clarify if interviews were conducted by trained enumerators and whether there was any quality control (e.g., supervisor checks, data audits).

10. The operationalization of psychological distress and financial indicators is clear and appropriate. However, grouping categories for logistic regression should be justified—for example, why >9 on depression is chosen as a cutoff.

11. The ethical procedures are appropriately described. However, mention how participant privacy was protected during face-to-face interviews in hospital settings (e.g., private rooms?).

12. Consider linking demographics to potential risk factors for psychological distress in the discussion section. Clarify the distribution of participants across the two hospitals.

13. Clarify figures for cancer stages in lines 185–186: Include the percentage for each stage (e.g., stage 1, 2, 3, 4) for completeness. Line 189: Rephrase to clarify—e.g., "The mean number of chemotherapy and radiotherapy cycles completed was 6.63 (SD = 5.01) and 15.71 (SD = 11.41), respectively." The data on multiple health facility use (lines 198–200) should be expanded with potential interpretation—what might this imply (e.g., care-seeking behavior, referrals, fragmented care)?

14. Add comparative reference to national or global averages for psychological distress among cancer patients to contextualize these findings. Briefly describe what the DASS-21 thresholds mean for each category (especially in figure captions or supplemental materials, lines 206-208).

15. Improve narrative clarity: Instead of repeating “In reference to...”, consider a tabular summary with narrative highlights for readability. Model II is mentioned multiple times, but results are only partially elaborated. Explicitly state which variables remained significant and how the strength of association changed after adjustment. It would be helpful to briefly define each model before presenting the results (e.g., "Model I includes unadjusted estimates, Model II includes adjusted variables", line 212 and so on).

16. Some sections are repetitive or overly descriptive (e.g., lines 264–275). Condense citations or summarize patterns with fewer studies. The explanation about family size not being significant (lines 293–295) is speculative—this should be clearly marked as a hypothesis, or replaced with a suggestion for future study. The claim that financial burden is not associated with distress (lines 313–319) is surprising and contrasts with global literature. This result needs: a more thorough explanation (e.g., potential limitations in measurement) and a discussion on possible underreporting or cultural resilience factors.

17. Reframe “psychosocial costs should not be one of the costs patients pay” (line 331). It’s powerful but could be more specific—e.g., "Psychological distress must be recognized and mitigated as part of holistic cancer care." Expand slightly on how the cancer control strategy (2024–2030) can support mental health—e.g., training counselors, routine distress screening, or including mental health in financial protection packages.

18. Improve the layout of the tables.

Reviewer #2: Although the manuscript is presented in an intelligible fasion and written in standard English, the manuscript is missing pertinent details such as if a particular sampling method will be used to recruit participants, if an inclusion and exclusion criteria will be implored, and how the sample size was determined.

**Do you want your identity to be public for this peer review?** For information about this choice, including consent withdrawal, please see our Privacy Policy

Reviewer #1: **Yes:** Gino A. Cabrera

Reviewer #2: **Yes:** Mah Wasi Asombang

---

## [Decision Letter · Decision Letter 1]

9 Oct 2025

PMEN-D-25-00335R1

Psychological distress and its associated factors among cancer patients in Nepal: a cross-sectional study

PLOS Mental Health

Dear <!--StartFragmentMr. Ankit Acharya<!--EndFragment,

Thank you for submitting your manuscript to PLOS Mental Health. After careful consideration, we feel that it has merit but does not fully meet PLOS Mental Health’s publication criteria as it currently stands. Therefore, we invite you to submit a revised version of the manuscript that addresses the points raised during the review process.

We look forward to receiving your revised manuscript.

Kind regards,

Laalithya Konduru

Academic Editor

PLOS Mental Health

Journal Requirements:

Reviewers' comments:

Reviewer's Responses to Questions

**Comments to the Author**

Reviewer #1: All comments have been addressed

Reviewer #3: (No Response)

publication criteria?

Reviewer #1: Yes

Reviewer #3: Partly

3. Has the statistical analysis been performed appropriately and rigorously?

Reviewer #1: Yes

Reviewer #3: Yes

4. Have the authors made all data underlying the findings in their manuscript fully available (please refer to the Data Availability Statement at the start of the manuscript PDF file)?

Reviewer #1: Yes

Reviewer #3: Yes

5. Is the manuscript presented in an intelligible fashion and written in standard English?

Reviewer #1: Yes

Reviewer #3: No

Reviewer #1: he manuscript addresses an important and under-researched area—psychological distress among cancer patients in Nepal. The study is relevant and timely, especially in low-resource settings. However, the manuscript requires:

1. Language polishing, improved coherence, and clarifications in methods and results to enhance its readability and scientific rigor.

2. More citations or statistics to support the claim that psychological distress is frequently overlooked.

3. Clarity in the gaps that this study fills.

4. Caution with interpretation, particularly with statistical association, causation should not be implied. Also, consider expanding on why no association was found—could it be due to measurement issues or coping mechanisms?

6. Recommendations are important but should be more concise and clearly actionable.

Reviewer #3: The abstract is wordy and has some awkward phrasing (“Patients at no cost should undergo through psychological distress” is unclear). It needs language refinement

The introduction is comprehensive but overburdened with detail (especially about radiotherapy and chemotherapy side effects). It risks losing focus on the rationale for the study. It must be streamlined.

The stated research gap is still vague. You note that distress is “least understood and often ignored,” but don’t explicitly show how prior Nepal-specific or South Asian evidence is lacking. More local or regional citations are needed.

The sampling description is complex. While two-stage random sampling is commendable, it’s not clear how representative the final sample of 262 is, given that 400 were originally recruited. Potential selection bias from those who declined DASS-21 should be acknowledged more strongly.

The sample size calculation is overly technical for the main text and doesn’t clearly link to the actual analytic sample. Why was a “two independent groups” formula used when this is a descriptive cross-sectional study? Simplification and better justification would help.

Training of enumerators (four hours on Zoom) seems minimal for sensitive psychological assessment. This limitation should be acknowledged.

Operational definitions are useful, but repeating cutoffs for DASS-21 in such detail is redundant. Please move to supplementary material.

The prevalence of distress is very high compared with global norms. This raises the question of whether cutoffs were too sensitive, or whether cultural/contextual factors inflated reporting. This issue is not discussed.

Tables are extremely dense and may overwhelm readers. Please move some to supplementary.

Reporting of missing data (e.g., why only 177 respondents for impoverishment) is buried. This needs to be clearer.

While you compare prevalence with global studies, the interpretation is somewhat superficial. Why is prevalence in Nepal higher? Structural issues (health system capacity, stigma, financial hardship, cultural perceptions of mental health) should be more deeply explored.

The null findings regarding OOPE, CATA, and impoverishment are glossed over. This is surprising given the strong theoretical link, and deserves a more thorough exploration (measurement error, timing of cost assessment, subsidies masking true burden, coping strategies, etc.).

The claim that unemployment “increased” after diagnosis is based on retrospective reporting, not longitudinal follow-up. So, causality is not justified.

Recommendations are too general (“Government must strongly implement…”) and not grounded in the study’s findings. More practical, targeted actions (e.g., integrating routine DASS-21 screening in oncology wards, training oncology nurses in basic counseling) need to be mentioned.

Some important limitations are omitted: Cross-sectional design prevents causal inference; possible social desirability bias in face-to-face interviews; use of self-report tool (DASS-21) without clinical validation in this sample; non-response bias from patients who refused/failed to complete DASS-21.

There are recurrent grammar and syntax issues (e.g., “Patients at no cost should undergo through psychological distress” is not correct English). Please consider getting your manuscript reviewed by native English speakers before submitting the revision.

Coherence between sections is uneven: the methods are very detailed, while the discussion feels shallow in parts.

**Do you want your identity to be public for this peer review?** For information about this choice, including consent withdrawal, please see our Privacy Policy

Reviewer #1: **Yes:** Gino A. Cabrera

Reviewer #3: No

---

## [Decision Letter · Decision Letter 2]

12 Dec 2025

PMEN-D-25-00335R2

Psychological distress and its associated factors among cancer patients in Nepal: a cross-sectional study

PLOS Mental Health

Dear Dr. Acharya,

Thank you for submitting your manuscript to PLOS Mental Health. After careful consideration, we feel that it has merit but does not fully meet PLOS Mental Health’s publication criteria as it currently stands. Therefore, we invite you to submit a revised version of the manuscript that addresses the points raised during the review process.

We look forward to receiving your revised manuscript.

Kind regards,

Laalithya Konduru

Academic Editor

PLOS Mental Health

Journal Requirements:

Reviewers' comments:

Reviewer's Responses to Questions

**Comments to the Author**

Reviewer #1: All comments have been addressed

Reviewer #3: All comments have been addressed

publication criteria?

Reviewer #1: Yes

Reviewer #3: Yes

3. Has the statistical analysis been performed appropriately and rigorously?

Reviewer #1: Yes

Reviewer #3: (No Response)

4. Have the authors made all data underlying the findings in their manuscript fully available (please refer to the Data Availability Statement at the start of the manuscript PDF file)?

Reviewer #1: Yes

Reviewer #3: No

5. Is the manuscript presented in an intelligible fashion and written in standard English?

Reviewer #1: Yes

Reviewer #3: Yes

Reviewer #1: Ensure consistency in reporting p-values (e.g., “p = 0.004” vs “p-value = 0.004”). The finding that economic variables (out-of-pocket expenses, catastrophic expenditure) were not significant warrants more discussion—particularly possible confounding or measurement limitations. The conclusion states that psychological distress is linked “primarily to sociodemographic and clinical factors rather than economic burden,” but the discussion does not adequately explain why economic indicators were not significant, given their relevance in LMIC settings. Consider comparing findings to regional or global studies to contextualize whether these levels are expected.

Reviewer #3: All previous comments have been addressed

**Do you want your identity to be public for this peer review?** For information about this choice, including consent withdrawal, please see our Privacy Policy

Reviewer #1: **Yes:** Gino A. Cabrera

Reviewer #3: No

---

## [Decision Letter · Decision Letter 3]

2 Jan 2026

PMEN-D-25-00335R3

Psychological distress and its associated factors among cancer patients in Nepal: a cross-sectional study

PLOS Mental Health

Dear Mr. Acharya,

Thank you for submitting your manuscript to PLOS Mental Health. After careful consideration, we feel that it has merit but does not fully meet PLOS Mental Health’s publication criteria as it currently stands. Therefore, we invite you to submit a revised version of the manuscript that addresses the points raised during the review process.

We look forward to receiving your revised manuscript.

Kind regards,

Laalithya Konduru

Academic Editor

PLOS Mental Health

Journal Requirements:

Reviewers' comments:

Reviewer's Responses to Questions

**Comments to the Author**

Reviewer #1: All comments have been addressed

publication criteria?

Reviewer #1: Yes

3. Has the statistical analysis been performed appropriately and rigorously?

Reviewer #1: Yes

4. Have the authors made all data underlying the findings in their manuscript fully available (please refer to the Data Availability Statement at the start of the manuscript PDF file)?

Reviewer #1: Yes

5. Is the manuscript presented in an intelligible fashion and written in standard English?

Reviewer #1: Yes

Reviewer #1:

1. Still needing stronger theoretical or conceptual framework.

2. Provide logical explanation on what guided the selection of sociodemographic, clinical, and economic variables. The study was conducted in two tertiary hospitals, which may overrepresent advanced-stage cancer cases and economically vulnerable patients, including rural or non-tertiary care settings.

3. Reconcile this: DASS-21 measures symptom severity, not clinical diagnoses and high prevalence rates (>65%) raise the possibility of measurement inflation due to somatic overlap with cancer symptoms. Discuss the possibility that cancer-related physical symptoms may artificially elevate DASS-21 scores.

4. Consider cultural norms, financial disclosure or family support systems to explain the lack of observed association.

**Do you want your identity to be public for this peer review?** For information about this choice, including consent withdrawal, please see our Privacy Policy

Reviewer #1: **Yes:** Gino A. Cabrera

---

## [Decision Letter · Decision Letter 4]

19 Feb 2026

Psychological distress and its associated factors among cancer patients in Nepal: a cross-sectional study

PMEN-D-25-00335R4

Dear Mr Acharya,

We are pleased to inform you that your manuscript 'Psychological distress and its associated factors among cancer patients in Nepal: a cross-sectional study' has been provisionally accepted for publication in PLOS Mental Health.

Best regards,

Laalithya Konduru

Academic Editor

PLOS Mental Health

Reviewer Comments (if any, and for reference):

Reviewer's Responses to Questions

**Comments to the Author**

Reviewer #3: All comments have been addressed

publication criteria?

Reviewer #3: Yes

3. Has the statistical analysis been performed appropriately and rigorously?

Reviewer #3: Yes

4. Have the authors made all data underlying the findings in their manuscript fully available (please refer to the Data Availability Statement at the start of the manuscript PDF file)?

Reviewer #3: Yes

5. Is the manuscript presented in an intelligible fashion and written in standard English?

Reviewer #3: Yes

Reviewer #3: (No Response)

**Do you want your identity to be public for this peer review?** For information about this choice, including consent withdrawal, please see our Privacy Policy

Reviewer #3: No
